# Peculiarities of the Phase Formation during Electroconsolidation of Al_2_O_3_–SiO_2_–ZrO_2_ Powders Mixtures

**DOI:** 10.3390/ma15176073

**Published:** 2022-09-01

**Authors:** Zbigniew Krzysiak, Edwin Gevorkyan, Volodymyr Nerubatskyi, Mirosław Rucki, Volodymyr Chyshkala, Jacek Caban, Tomasz Mazur

**Affiliations:** 1Faculty of Production Engineering, University of Life Sciences in Lublin, Głęboka 28, 20-612 Lublin, Poland; 2Wagon Engineering and Production Quality, Ukraine State University of Railway Transport, 7 Feuerbach Sq., 61010 Kharkiv, Ukraine; 3Institute of Mechanical Science, Vilnius Gediminas Technical University, J. Basanaviciaus str. 28, LT-03224 Vilnius, Lithuania; 4Department of Reactor Engineering Materials and Physical Technologies, V. N. Karazin Kharkiv National University, 4 Svobody Sq., 61022 Kharkiv, Ukraine; 5Sieć Badawcza Łukasiewicz, Instytut Technologii Ekspoatacji w Radomiu, ul. K.Pułaskiego 6/10, 26-600 Radom, Poland; 6Faculty of Mechanical Engineering, Lublin University of Technology, Nadbystrzycka 36, 20-618 Lublin, Poland; 7Faculty of Mechanical Engineering, Kazimierz Pulaski University of Technology and Humanities in Radom, Stasieckiego 54, 26-600 Radom, Poland

**Keywords:** ceramic composite, Al_2_O_3_–SiO_2_–ZrO_2_, electroconsolidation, mullite, corundum, glassy phase, refractory, XRD, zirconia, elasticity

## Abstract

This paper is devoted to the sintering process of Al_2_O_3_–SiO_2_–ZrO_2_ ceramics. The studied method was electroconsolidation with directly applied electric current. This method provides substantial improvements to the mechanical properties of the sintered samples compared to the traditional sintering in the air. The research covered elemental and phase analysis of the samples, which revealed phase transition of high-alumina solid solutions into mullite and corundum. Zirconia was represented mainly by tetragonal phase, but monoclinic phase was present, too. Electroconsolidation enabled samples to reach a density of 3.0 g/cm^3^ at 1300 °C, while the sample prepared by traditional sintering method obtained it only at 1700 °C. For the composite Al_2_O_3_—20 wt.% SiO_2_—10 wt.% ZrO_2_ fabricated by electroconsolidation, it was demonstrated that fracture toughness was higher by 20–30%, and hardness was higher by 15–20% compared to that of samples sintered traditionally. Similarly, the samples fabricated by electroconsolidation exhibited elastic modulus *E* higher by 15–20%. The hypothesis was proposed that the difference in mechanical and physical properties could be attributed to the peculiarities of phase formation processes during electroconsolidation.

## 1. Introduction

Sintering is one of the most powerful technologies to obtain ceramics and cermets of starting powders [1]. Alumina Al_2_O_3_ is one of the ceramic materials widely used in industrial applications [2,3,4,5]. Among refractories, the Al_2_O_3_–SiO_2_–ZrO_2_ system is an important category, suitable for many applications, such as steel and glass manufacturing industries, due to their high refractory characteristics, chemical resistance, toughness, high-creep resistance and adequate thermo-mechanical performance [6]. Its composition is based on corundum, mullite and baddeleyite structures and consists of variable amounts of vitreous phase, zirconia polymorphs, raw and calcined kaolins, alumina of different particle sizes, as well as zirconium compounds, including zirconia–mullite composites [7]. It is important to consider that the advantageous properties may vary dependent on the amount of mullite and zirconia inclusions, since the percentage of these inclusions in the matrix has a direct impact on the efficiency of the toughening mechanisms [8]. The crystallization and phase transformation processes have a critical effect on the thermo-mechanical performance and on the fracture behavior of silica refractories [9], while a phase composition, in turn, depends on the sintering process parameters [10].

During the bonding reaction, the zircon dissociation and mullite formation is initiated at ca. 1400 °C and completed by ca. 1500 °C. It is accompanied by a radial decomposition of zircon propagating from the outer surface to the grain center, which promotes a rapid crystallization of zirconia and leads to the formation of amorphous silica [11]. Thus, sintering of alumina–mullite–zirconia refractories is usually carried out in the 1400–1600 °C range, with prolonged soaking, to obtain the desired phase composition through several phase transformations [7]:-decomposition of minerals, dehydroxilation of kaolinite at ca. 500 °C and further transition into mullite and silica at temperatures over 1000 °C;-disappearance of quartz after its incorporation in mullite or the liquid, usually above 1300 °C;-at temperatures over 1400 °C takes place a dissociation of zircon into zirconia, which upon cooling is transformed from the tetragonal to the monoclinic form.

Zirconia inclusions due to their phase transformation can build up advantageous compressive stresses in the matrix, and also reduce the average size of the alumina phase grains [12]. In order to increase the mullite amount in alumina ceramics, either proper mullite-containing material may be added, or proper silicate-based material may be used as additives promoting formation of mullite and residual crystalline phases [13]. Most of the research reports address the issue of initial mixtures, focusing on the micron and submicron size powders, further sintered usually into the air environment. For instance, Medvedovski studied not only alumina ceramics based on Al_2_O_3_–SiO_2_ system with additions of earth-alkali silicates or borosilicates, but also additives stabilizing ZrO_2_ and promoting mullite formation [14]. The author demonstrated that the materials had uniform fine-crystalline microstructures, with remarkably improved physical properties, as well as high wear resistance. In other research, different binder compositions were tested, namely, quartz/α–Al_2_O_3_ and kaolin/α–Al_2_O_3_, along with ZrSiO_4_ addition [15]. Thermal shock resistance was analyzed for the materials of different composition, e.g., densely sintered alumina of high-purity with grains smaller than 5 μm, alumina-based composite reinforced with 30 vol.% of silicon carbide whiskers, and titanium-based cermet [16]. Composites with different Al_2_O_3_/SiO_2_ ratios were investigated in respect to the microstructure and mechanical properties, as well as hot-pressed alumina–mullite–zirconia–silicon carbide composites [17]. The authors reported excellent combined properties achieved for SiC 20 vol.% content sintered at temperature of 1530 °C, with Vickers hardness of 17.5 GPa, fracture toughness 12.3 MPa m^1/2^, and flexural strength 970 MPa. In [18] it was reported that zircon content contributed to the increase in strength and modulus values due to a significant improvement in densification and a reduction in porosity by filling of voids.

The effect of the phase content and morphology of initial powder particles has been considered by researchers, too [19,20]. In the paper [21], it is reported that a fine-grain matrix was formed due to the dissociation of zircon and silica reactions, as well as bonding of the finer fractions of alumina and mullite powders, and the products of zircon dissociation, where a liquid phase is most likely involved. In general, the researchers are focused on the initial phase components and sintering temperatures.

Above-mentioned papers indicated that additions of zirconia could be a promising research direction, especially when considering the formation of a new phase under different sintering conditions. Further improvement of the properties of a ceramic composite for cutting tool application is still possible. The present work is focused on the examination of the structure formation during the sintering process, in the phase composition context, and subsequent physical and mechanical characteristics. In the research, four novel aspects can be named. Among available sintering methods, a novel hot-pressing field-assisted technique was chosen. Next, the investigations were made on the powder mixtures of novel compositions. Moreover, the powders were made using original, innovative technologies. Finally, these three unusual factors allowed for obtaining a unique structure containing mullite, as it is demonstrated and discussed below.

## 2. Materials and Methods

### 2.1. Initial Powders

Initial Al_2_O_3_–SiO_2_–ZrO*_2_* nanopowders used for sintering are shown in Figure 1. It is seen that the grain sizes are very small and uniformly mixed together.

In the experiments, nanopowders ZrO_2_ partially stabilized by 5 wt.% Y_2_O_3_ were used. An SEM image of this powder is presented in Figure 2, while in Figure 3, there is an SEM image of α–Al_2_O_3_ nanopowder. Both materials were delivered by NANOE (Ballainvilliers, France). The homogenous mixture of these powders was prepared using a planetary mill Pulverisette-6 (Fritsch GmbH, Idar-Oberstein, Germany) with a dry method. Then, an isopropyl alcohol was added, and the powders were mixed during 2 h. Rotational speed of the planetary disc was 160 rpm. Afterwards, the powders were dried up. Large agglomerates, seen in Figure 2, were destroyed and not found in the final mixture of powders.

Alumina powder consisted mainly of the α–Al_2_O_3_, but other phases were also present, apparently. From the XRD graphs shown in Figure 4, it can be concluded that the powder contained some *γ*– and *θ*–Al_2_O_3_, causing a difference from reference curve 2 to the standard α–Al_2_O_3_ (99.5% pure).

Nanopowders SiO_2_ delivered by Guandzou Hongwu Material Technology Co. Ltd. (Guangzhou, China) were used for preparation of Al_2_O_3_–SiO_2_–ZrO_2_ powders mixture. Their morphology can be seen in Figure 5.

### 2.2. Samples Preparation

After obtaining a mixture, nanopowders underwent calcination at 800 °C during 1 h, and then were sintered using a patented hot-pressing device, described in detail elsewhere [22]. The sample was placed into the vacuum chamber and, after being heated by the electric current, underent mechanical pressure to form a dense, solid structure. The process was called electroconsolidation to distinguish it from the Spark Plasma Sintering (SPS) and Field Activated methods. We preferred to avoid a SPS nomenclature before ensuring that plasma really appeared and played any role in the consolidation process. In fact, an alternating current of 3000 A was directly applied to the graphite molds, and the released heat caused consolidation of the pressured powder. Thus, the term ‘electroconsolidation’ emphasizes the main role of the electric current in the powder consolidation process.

Electroconsolidation was performed in the vacuum of 10^−2^ Pa at different temperatures from 1300 °C up to 1700 °C. The initial investigations allowed for setting some limits on other parameters. In particular, it was found that better properties of the sintered material were obtained at mechanical pressure 35 MPa and heating rate 250 °C/min. As-obtained samples of diameter 12 mm and thickness 5 mm were polished with grinding-polishing unit Struers (Ballerup, Denmark), using diamond polishing pastes gradually down to 1 μm.

For the comparison of mechanical properties, samples were prepared also by traditional sintering. The powders in proportion Al_2_O_3_—20 wt.% SiO_2_—10 wt.% ZrO_2_ were mixed together in the mill for 5 h and then pressed in the steel molds under 100 MPa. These samples were then sintered in the furnaces NaberTherm (Lilienthal, Germany) at similar temperatures from 1300 °C up to 1700 °C during 1 h. It was noted that at 1300 °C porosity was as high as 15%, while after sintering at 1700 °C it was 5%.

### 2.3. Tests and Measurements

The structure of as-prepared samples was examined with a field emission microscopy (FEM), Nova NanoSEM (made by FEI, Hillsboro, OR, USA) with Secondary Electrons/Backscattered Electrons in-lens detection and beam deceleration ensuring resolution of 1.8 nm at 3 kV. Other device was Quanta 200 3D (made by FEI, Hillsboro, OR, USA), a dual-beam scanning electron microscope of resolution 50 nm at 30 kV. Dimensions of image areas were as follows: 1 × 1 μm, 2.5 × 2.5 μm, and 5 × 5 μm.

Mappings of the elemental distribution in the sintered samples were obtained using a computerized color cathodoluminescent (CCL) attachment on a scanning electron microscope. The elemental analysis of the samples was performed with a SEM unit LEO1455 VP (ZEISS company, Berlin, Germany).

The X-ray analysis was performed with a diffractometer Shimadzu XRD-6000 (Shimadzu company, Wolverton, UK) with a graphite monochromator counter and CuKα radiation of λ = 1.54187 Å. X-ray tube current was 30 mA and voltage 40 kV, with stability guaranteed within ±0.01%. The speed of continuous scanning θ–2θ was 1.2°/min, in the range from 5.0 to 100.0° with step 0.02°, while the sample did not rotate. To perform the phase analysis, the ASTM (American Society for Testing Materials) database was applied.

The microhardness was measured with an automatic AFFRI Hardness Tester DM8 produced by Scicron (Thanon Vacharaphol, Thailand). The indenter was in form of a Vickers pyramid with an angle of α = 136°. The microhardness *HV* was calculated from the following equation [23]:(1)$$HV=\frac{k\cdot P}{{\left(2\cdot a\right)}^{2}},$$
where *P*—load on the indenter, in our study *P* = 10 N, 2·*a*—an average length of the indentation diagonal [μm], *k*—is the factor taken for a Vickers pyramid *k* = 1.854.

To assess the crack resistance of the obtained samples, fracture toughness was calculated from the following equation [24]:(2)$$K_{IC}=0.016\cdot {\left(\frac{l}{a}\right)}^{-0.5}\cdot {\left(\frac{H_{V}}{E\cdot Ф}\right)}^{-0.4}\cdot \frac{H_{V}\cdot a^{0.5}}{Ф},$$
under the condition of ratio:(3)$$0.25\leq \frac{l}{a}\leq 2.5,$$
where *E*—an elastic modulus [GPa], *Ф*—constant [-] that took here the value *Ф* ≈ 3, *l*—a crack length measured from the indentation corner [μm], *a*—half of the length of the indentation diagonal [μm].

The elastic modulus *E* was measured for the samples prepared at different temperatures by both electroconsolidation and traditional sintering methods. It was defined according to the methodology ASTM E1876-01 [25] using the device Buzz-o-Sonic (BuzzMac International, LLC, Milwaukee, WI, USA). The modulus was calculated from the formula derived from [25], as follows:(4)$$E=0.9465\rho f2L^{4}\left(\frac{T_{1}}{t^{2}}\right),$$
where *ρ*—density [g/cm^3^], *f*—sound propagation frequency [Hz], *L*—sample length [mm], *T*_1_—a correction factor dependent on the size of the sample, *t*—sample thickness [mm].

## 3. Theoretical Background

Generally, mullite–zirconia composites can be produced by various methods, such as direct powder coating, mullite–zirconia sintering, alumina–silica–zirconia reaction sintering; reaction sintering of alumina, kaolinite and zircon; spark plasma sintering, a directed laser floating zone, a standard curing method, sol-gel and reaction sintering. Monolithic unmodified mullite ceramics do not always provide the required mechanical characteristics, including elasticity, bending strength, or fracture toughness. The addition of zirconia, often stabilized with yttria, increase hardness, wear resistance, and the elastic modulus close to the softening temperature [26,27].

In this context, a composition of the tested samples was chosen in a proportion that made it possible to perform the solid phase exchange interaction [28,29]:(5)$$3{\mathrm{Al}}_{2}O_{3}+2{\mathrm{ZrO}}_{2}\cdot {\mathrm{SiO}}_{2}=3{\mathrm{Al}}_{2}O_{3}\cdot 2{\mathrm{SiO}}_{2}+2{\mathrm{ZrO}}_{2}.$$

According to the available published data [30], reaction expressed by Equation (5) occurs at the temperatures above 1342 °C and is accompanied by the increase in volume of almost 12%. Thus, the proportion of components by weight percentage of 70% of Al_2_O_3_, 20% of SiO_2_, and 10% of ZrO_2_, allowed for obtaining calculated composition at 1342 °C in following proportions: Al_2_O_3_—20.68 wt.%, ZrO_2_∙SiO_2_—24.80 wt.%, and 3Al_2_O_3_∙2SiO_2_—54.52 wt.%.

In fact, during the reaction (5), the phase composition tends to reach a theoretically possible proportion with 16.66 wt.% of ZrO_2_ and 83.34 wt.% of 3Al_2_O_3_∙2SiO_2_. In other words, all the zircon synthesized before would react with the corundum, forming a new ZrO_2_ and additional mullite. It is difficult to investigate in detail phase transformations of the thermally aged heterogeneous materials due to the large number of significant influencing factors that should be considered. Nevertheless, the main conclusion may be drawn from the experimental results and generalizations made according to the presented theoretical background. In particular, all the temperatures of the reactions do not exceed 1386 °C, which provides conditions for the phase decomposition of high-alumina solid solutions into mullite and corundum [30]. That is why the latter appeared in the sintered samples. This conclusion is supported by the fact of thermodynamic stability of corundum in combination with zirconium and mullite, according to the low-temperature triangulation of ZrO_2_–Al_2_O_3_–SiO_2_ system. Thus, it can be written as follows:(6)$$2{\mathrm{ZrSiO}}_{4}+\left(3+x\right)\cdot {\mathrm{Al}}_{2}O_{3}=x\cdot {\mathrm{Al}}_{2}O_{3}+2{\mathrm{ZrO}}_{2}+3{\mathrm{Al}}_{2}O_{3}\cdot 2{\mathrm{SiO}}_{2}.$$

The literature provides enough information on phase formation in the Al_2_O_3_–ZrO_2_–SiO_2_ system. In particular, the classical phase diagram of Al_2_O_3_–ZrO_2_–SiO_2_ can be found in [31], and the updated one version using the Nuclea database and the available experimental data was published in [32]. The phase diagram and the Equation (6) allow for the exact calculation of the initial components to obtain the required phase composition after transformations of the sintering process. In the present study, the initial powder materials were portioned in a way that provides the final phase composition with proportions 60 wt.% of Al_2_O_3_, 25 wt.% of mullite, and 15 wt.% of ZrO_2_. Moreover, previous studies on alumina–zirconia–mullite composites, as well as research on sintering of alumina and zirconia, reported better sintering ability and enhanced properties of this composite in the abovementioned proportion [33,34,35].

## 4. Results and Discussion

### 4.1. Phase Composition and Microstructure

Results of XRD analysis of the samples sintered at different temperatures are shown in Figure 6. The sample #1 was fabricated at 1300 °C, #2 at 1400 °C, #3 at 1500 °C, and #4 at 1700 °C. Peaks correspond with mullite (M), corundum (C), and zirconia (Z), respectively. Additionally, Z_m_ stands for monoclinic phase of ZrO_2_.

XRD data presented in Figure 6 revealed that the samples consisted of mullite, corundum and zirconia, including its monoclinic phase. The graphs appear quite similar for the samples #1–3, hence, changes of sintering temperature in the range 1300 °C up to 1500 °C did not affect the phase composition of the ceramics. However, the diagram for sample #4 sintered at 1700 °C looks different. In particular, peaks representing monoclinic ZrO_2_ disappeared, which indicated that the monoclinic phase was not formed during sintering process at 1700 °C.

Notably, the sintered samples exhibited some inhomogeneity, with clear agglomerations. Thus, an area with visible uneven distribution of inclusions was chosen for the analysis. Figure 7, Figure 8, Figure 9 and Figure 10 indicate points on sample #4, where the elemental analysis was performed, together with corresponding spectra. Respective percentage of the elements in each point are presented in Table 1, Table 2, Table 3 and Table 4. Elemental analysis of other samples sintered at lower temperatures did not show any significant differences.

Figure 7 and Table 1 reveal that in spectrum 1, alumina was dominant with small amount of mullite. In contrast, Figure 8 spectrum 2 indicates the main phase of zirconia among elements presented in Table 2. At the same time, Figure 9 in spectrum 3 shows mainly mullite and zirconia. It may be deduced that zirconia is mainly represented by its tetragonal phase with monoclinic inclusions revealed by WRD analysis. Graphs in Figure 6 suggest that in sample #4 sintered at 1700 °C, monoclinic phase was almost absent. It can be assumed, however, that the presence of small amount of monoclinic zirconia would not significantly affect physical and mechanical properties of the composite.

Figure 11 presents the structure in point 4 of the sample #4. The composite here consisted of mullite, ca. 60%, and zirconia. There can be seen almost uniform prism-like grains, quite steadily distributed throughout the composite. Grain size is close to 1 μm, which can be considered a fine-dispersed microstructure.

These characteristics of the obtained microstructure, especially in the context of the observed phase transformation, determined physical and mechanical properties of the tested composite.

### 4.2. Physical and Mechanical Properties

Since the examined ceramic material is designed for tool applications, analysis of the physical and mechanical properties was limited to the density, fracture toughness, and microhardness. The main topic of the comparative analysis is focused on the difference between traditional sintering method and the novel one with all its peculiarities. The analysis below demonstrates that the proposed novelties are advantageous compared to the traditional sintering from the perspective of the analyzed mechanical properties.

It would be expected that densification is more effective at higher temperatures. The experiments confirmed an almost proportional increase in density with increasing sintering temperatures, which is seen in the graph in Figure 12. Comparison of the traditional sintering method (line 1) with the novel electroconsolidation (line 2) proves undoubted advantage of the latter method.

Electroconsolidation enabled samples to reach density of 3.0 g/cm^3^ at 1300 °C, while the sample prepared by traditional sintering method obtained it at 1700 °C, as it can be seen in Figure 12. In turn, electroconsolidation at 1700 °C made it possible to reach almost 3.5 g/cm^3^. The research demonstrated that the material of high mechanical properties feasible for cutting tool applications could be obtained at 1300 °C. From the theoretical and experimental diagrams [36], it is suggested that the minimal temperature for this material was 1750 °C. The difference could be attributed to the peculiarities of the electroconsolidation method.

On the one hand, that means higher energy efficiency of the electroconsolidation, and on the other hand, that ensured more fine-dispersed grain sizes, since elevated temperatures promote grain growth. As it is widely known, fine-grained structure is an enhancing factor and is more advantageous from the practical application perspective [37]. Moreover, extensive grain growth at higher temperatures was disadvantageous since it prevented from full densification of the ceramics.

As a result, fracture toughness of the ceramics after electroconsolidation was higher than that after traditional sintering. Respective curves are shown in Figure 13. Noteworthy, fracture toughness *K_IC_* of ca. 5.9 MPa∙m^½^ was obtained by electroconsolidation at 1300 °C, while for the traditional sintering it was a maximal possible value obtainable at 1500 °C. In both cases, however, maximal fracture toughness was reached at 1500 °C, and its value decreased at higher sintering temperatures, but *K_IC_* of the samples obtained by electroconsolidation was always by 20–30% higher. Similar trends are clearly seen in the graphs of hardness shown in Figure 14, with a shift of maximal value to the temperature of 1600 °C in the case of traditional sintering method.

It should be noted that the highest fracture toughness and microhardness were not obtained for the smallest porosity reached at 1700 °C. Decrease of the microhardness and fracture toughness sintered at 1700 °C can be explained by the decrease of the monoclinic phase of ZrO_2_ and the appearance of its amorphous phase under the influence of high temperatures. Moreover, the average grain sizes increased which, lead to weakening of the sintered material.

Figure 15 presents the plots of elastic modulus *E* measured for the tested Al_2_O_3_—20 wt.% SiO_2_—10 wt.% ZrO_2_ samples after different numbers of cyclic thermal loads. Two respective curves represent the samples traditionally sintered, and the ones fabricated by electroconsolidation method.

The samples prepared at 1400 °C by traditional sintering method and electroconsolidation underwent thermal loads. They were heated from 20 °C up to 1000 °C, kept in this temperature for 30 min and then cooled down to the room temperature. It should be noted that the phases formed during electroconsolidation and their microstructures provided much higher elastic modulus than that of traditionally sintered samples of the same material. In general, *E* was higher by 15–20% for the samples fabricated by electroconsolidation, with significant increase by 10 GPa after 6 thermal load cycles. In contrast, elastic modulus gradually decreased for the traditionally sintered sample, from 153 down to 140 GPa.

In the performed research, it was difficult to determine the effect of uneven distribution on the components in the sintered samples. However, it seems rather obvious that agglomerates might have weakened the macroscopic mechanical characteristics. Thus, in further research it is planned to prolong mixture preparation and to apply special methods to crush the agglomerates.

The presented above results confirmed the advantages of the electroconsolidation method over traditional sintering in air. The enhanced properties can be attributed to the phase formation processes, especially to the appearance of mullite. Large amounts of the mullite phase increased ductility of the composite and its strength. Moreover, the partial transformation of the tetragonal zirconia into monoclinic phase supported the enhancement.

## 5. Conclusions

The presented results demonstrated improvement of the mechanical properties of Al_2_O_3_–SiO_2_–ZrO_2_ ceramics fabricated using the electroconsolidation method, as compared to the traditional sintering in the air. Hypothetically, the difference could be attributed to the peculiarities of phase formation promoted by the processes that take place during electroconsolidation. First of all, the process provided increased density of the composite and prevented grains from growth due to lower temperatures and shorter time of sintering. Moreover, the electroconsolidation provided stronger bonds between phases of the composite, which resulted with increased fracture toughness.

The phase analysis showed that during electroconsolidation, formation of dense mullite areas took place, significantly different from typical prism-like shape. The difference included pin-like grains at the boundaries with mullite grains, as well as monoclinic zirconia inclusions. It can be concluded that these effects contributed to the improvement of the composite properties after electroconsolidation. In particular, fracture toughness was 20–30% higher, and hardness higher by 15–20% compared to that of samples fabricated by traditional sintering in the air environment. Similarly, elastic modulus *E* after heavy thermal cyclic loads was higher by 15–20% for the samples fabricated by electroconsolidation.

It can be also assumed that further improvement of the mechanical characteristics of the ceramic composites could be achieved through better preparation of the initial powders. In particular, homogeneity of the powder mixture can be further improved, and its composition can be further optimized.

## Figures and Tables

**Figure 1 materials-15-06073-f001:**
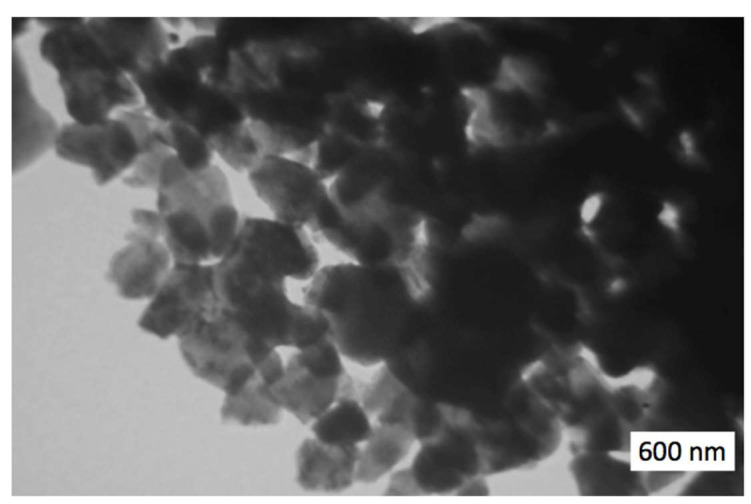
TEM image of the initial powder Al_2_O_3_—20 wt.% SiO_2_—10 wt.% ZrO_2_.

**Figure 2 materials-15-06073-f002:**
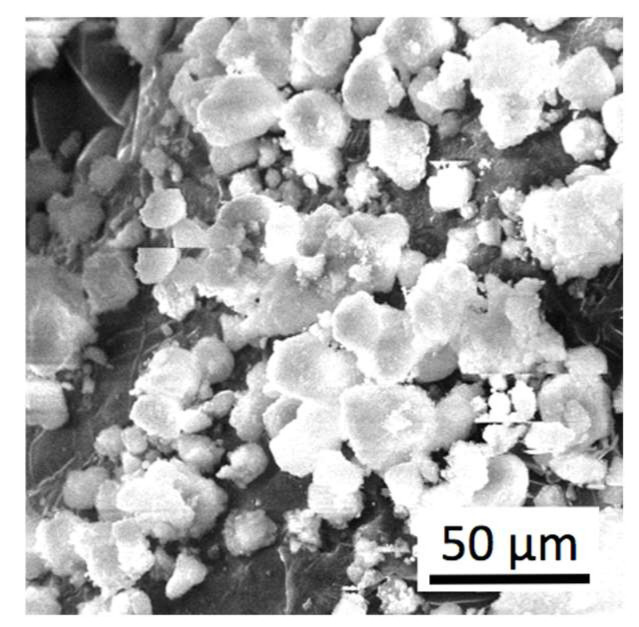
SEM image of ZrO_2_—5 wt.% Y_2_O_3_ nanopowder.

**Figure 3 materials-15-06073-f003:**
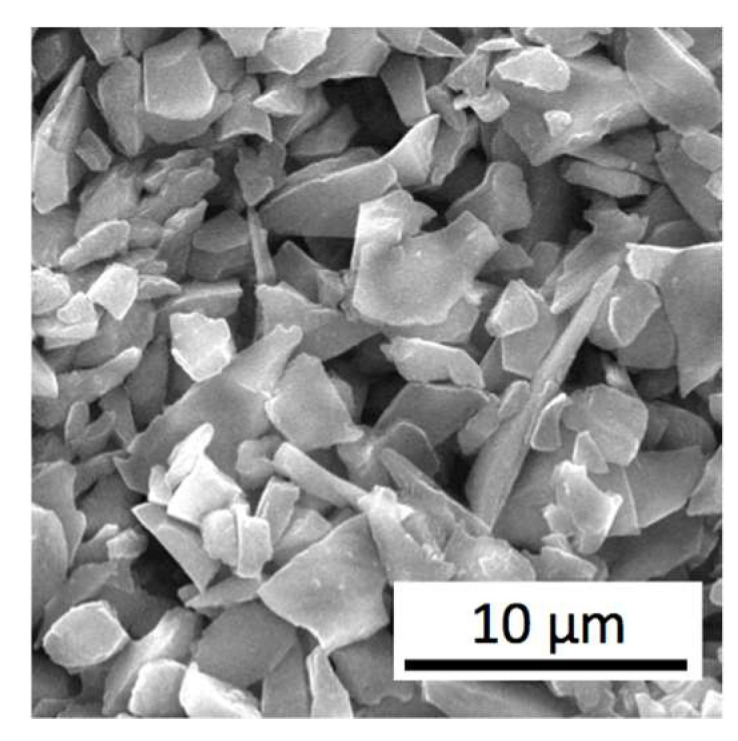
SEM image of α–Al_2_O_3_ nanopowder.

**Figure 4 materials-15-06073-f004:**
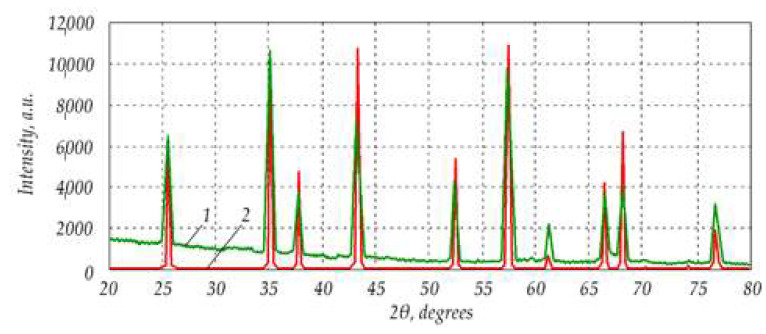
XRD analysis of nanopowders: 1—analysed Al_2_O_3_ powder; 2—standard α–Al_2_O_3_ phase.

**Figure 5 materials-15-06073-f005:**
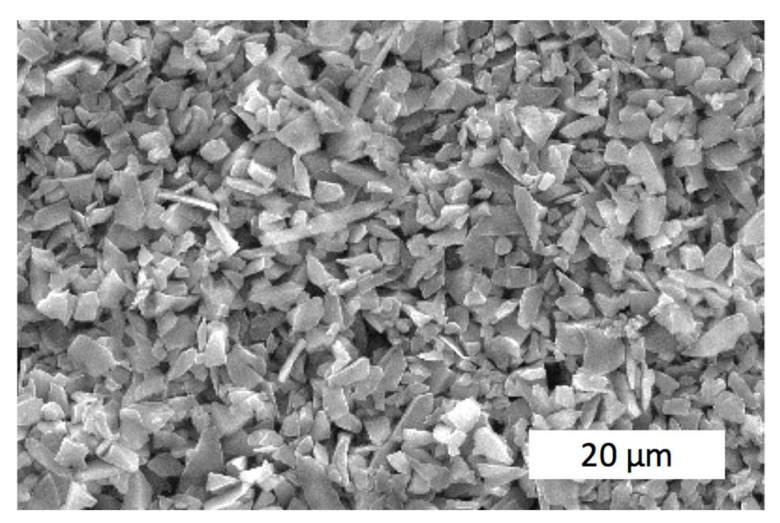
SEM image of nanopowders SiO_2_.

**Figure 6 materials-15-06073-f006:**
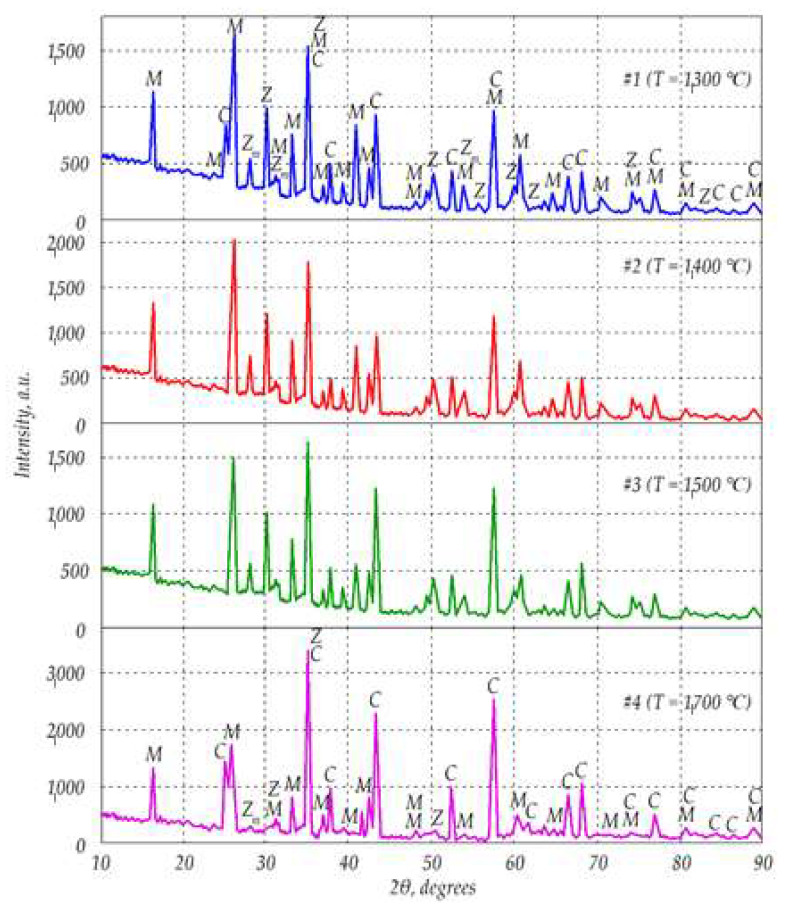
Diffractograms of the mullite-corundum ceramics sintered at different temperatures. *M*—mullite (3Al_2_O_3_∙2SiO_2_), *C*—corundum (α–Al_2_O_3_), *Z*—zirconia (ZrO_2_ tetragonal), and *Z_m_*—zirconia (ZrO_2_ monoclinic).

**Figure 7 materials-15-06073-f007:**
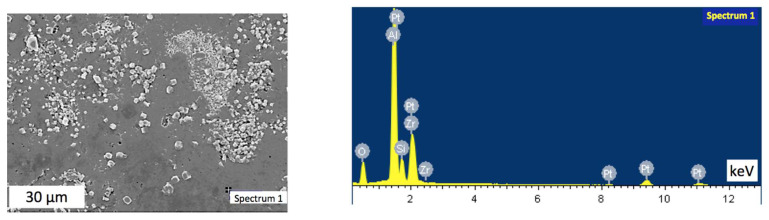
Elemental analysis in point 1 of the sample #4.

**Figure 8 materials-15-06073-f008:**
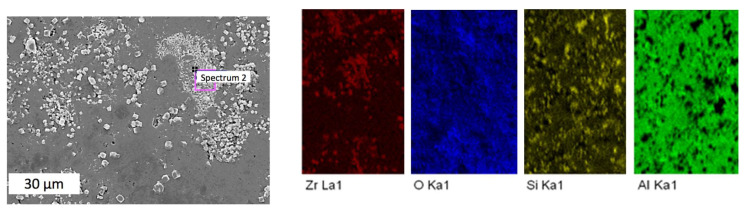
Elemental analysis in point 2 of the sample #4.

**Figure 9 materials-15-06073-f009:**
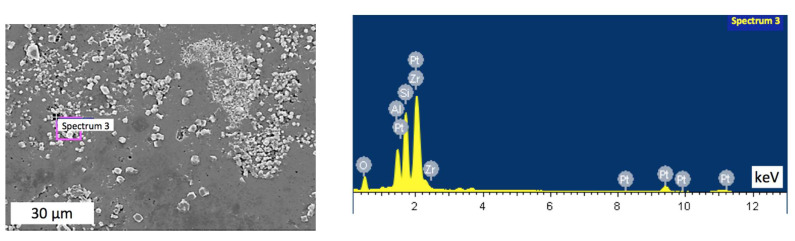
Elemental analysis in point 3 of the sample #4.

**Figure 10 materials-15-06073-f010:**
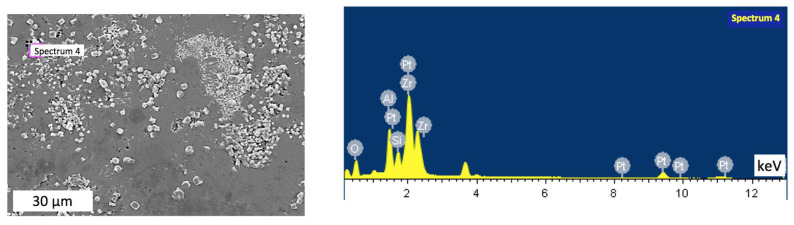
Elemental analysis in point 4 of the sample #4.

**Figure 11 materials-15-06073-f011:**
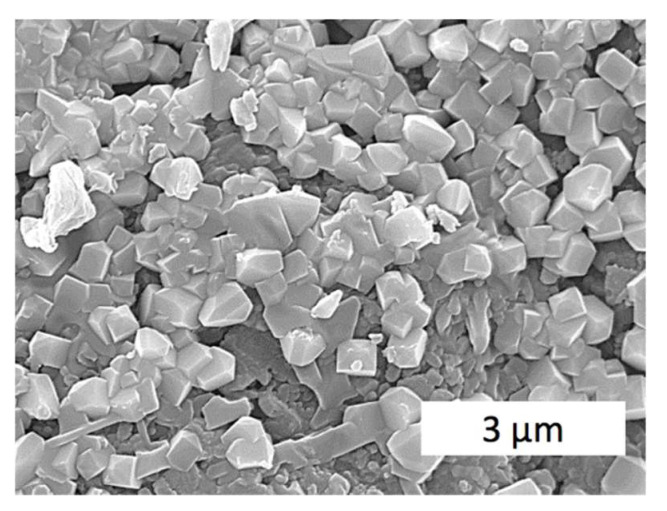
SEM image of the composite Al_2_O_3_—20 wt.% SiO_2_—10 wt.% ZrO_2_ fracture. The electroconsolidation was performed at 1700 °C, under pressure 30 MPa during 3 min.

**Figure 12 materials-15-06073-f012:**
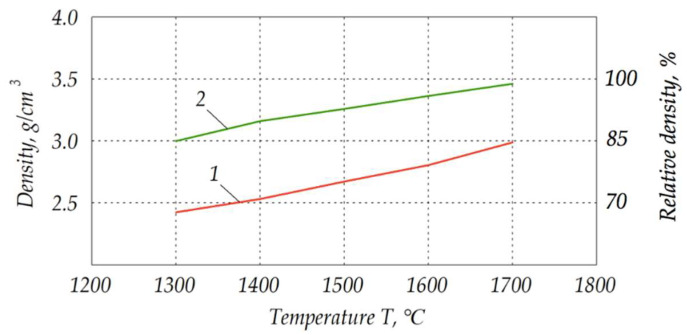
Density of the tested composites Al_2_O_3_—20 wt.% SiO_2_—10 wt.% ZrO_2_ prepared at different temperatures: *1*—by traditional sintering method, *2*—by electroconsolidation.

**Figure 13 materials-15-06073-f013:**
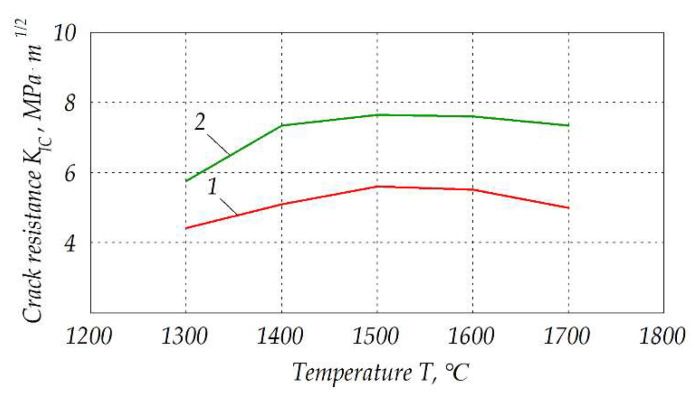
Fracture toughness of the composite Al_2_O_3_—20 wt.% SiO_2_—10 wt.% ZrO_2_ sintered at different temperatures: *1*—traditional sintering, *2*—electroconsolidation.

**Figure 14 materials-15-06073-f014:**
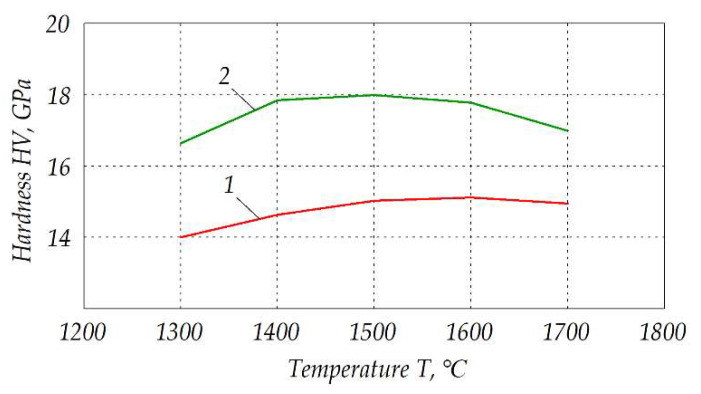
Microhardness of the composite Al_2_O_3_—20 wt.% SiO_2_—10 wt.% ZrO_2_ sintered at different temperatures: *1*—traditional sintering, *2*—electroconsolidation.

**Figure 15 materials-15-06073-f015:**
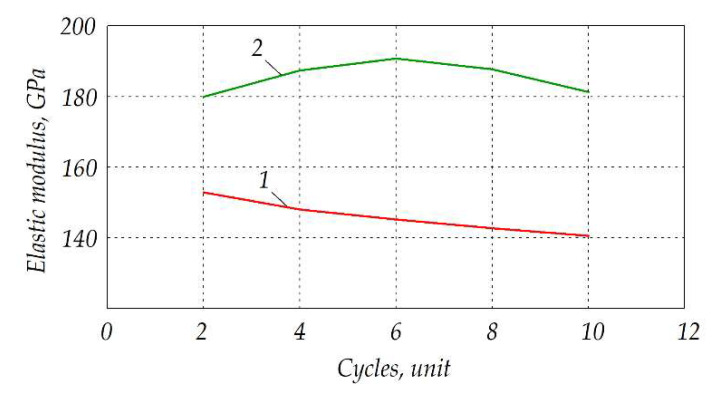
Elastic modulus *E* after heating cycles for Al_2_O_3_—20 wt.% SiO_2_—10 wt.% ZrO_2_ composite samples fabricated at 1400 °C by: *1*—traditional sintering, *2*—electroconsolidation.

**Table 1 materials-15-06073-t001:** Elemental proportions in point 1 of the sample #4.

Element	Weight, %	Atomic, %
O	39.20	56.95
Al	38.97	33.57
Si	6.85	5.67
Zr	14.97	3.81
Total	100.00	100.00

**Table 2 materials-15-06073-t002:** Elemental proportions in point 2 of the sample #4.

Element	Weight, %	Atomic, %
O	31.03	68.68
Al	0.24	0.31
Si	4.95	6.25
Zr	63.77	24.76
Total	100.00	

**Table 3 materials-15-06073-t003:** Elemental proportions in point 3 of the sample #4.

Element	Weight, %	Atomic, %
O	35.93	62.75
Al	8.20	8.49
Si	16.91	16.82
Zr	38.95	11.93
Total	100.00	

**Table 4 materials-15-06073-t004:** Elemental proportions in point 4 of the sample #4.

Element	Weight, %	Atomic, %
O	47.25	73.89
Al	12.40	11.50
Si	5.73	5.11
Zr	34.62	9.50
Result	100.00	

## Data Availability

Data available on request due to privacy restrictions.

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
