# Peer review of "Peculiarities of the Phase Formation during Electroconsolidation of Al2O3–SiO2–ZrO2 Powders Mixtures"

_materials, 2022, doi:10.3390/ma15176073_

Round 1

Reviewer 1 Report

The manuscript, “Peculiarities of the phase formation during electroconsolidation of Al2O3-SiO2-ZrO2 powders mixture”, is interesting, however, there are some issues that must be addressed:

1.       In title, powders mixture should be changed to powder mixtures.

2.       The English language should be revised.

3.       The author claimed that electroconsolidation sintering is different from spark plasma sintering (SPS) and field assisted sintering technology (FAST) as there was no observed plasma in during their sintering process. In my opinion, the plasma is observed for conducting materials such as sintering of metallic powders. For non-conducting ceramics, the plasma will not be formed. Non-conducting ceramics powder sintered with SPS and FAST were also reported.  Therefore, the author’s claim is not appropriate.

4.       Did author observed any carbon diffusion in their samples after sintering?

5.       If any method is used to protect the carbon diffusion, please mention it in the manuscript.

6.       What is the dimension of the sintered sample?

7.       How much the load is applied during hardness measurement?

8.       In Fig. 8, please remove mapping and include elemental spectrum. In mapping there is no Zr element.

9.       In fig. 10, the micrograph shows spectrum 3, which should be spectrum 4. Please correct it.

10.   Fig. 7-10, all images are for 1700 oC sintered sample. Why different positions have different elemental proportion? Either the sample is not well sintered or mixing of powders is not homogeneous.

11.   Figure 11 looks like fractured surface, please clarify it. Also, in figure caption, the hot-pressed at 1700 oC is mentioned, which should be changed to electroconsolidated.

12.   The author mentioned density of 1300 oC sintered sample but there were no data for other samples.

13.   There is no micrographs and elemental analysis of other samples sintered at other temperatures.

14.   Comparison is not clear; there is no other data for hot pressed samples, such as XRD and elemental analysis.

Overall, the manuscript will be much better, if other data can be supplied as supplementary or in the manuscript.  Fig. 2 and 3 have duplicate figures and please remove the duplicate ones.

Author Response

Dear Editors and Reviewers,

Thank you very much for the thorough review and valuable suggestions. We addressed them all, and marked in red all the alterations in the revised paper. The detailed responses are in the attached file.

Best regards

Reviewer 2 Report

Fig 1 - should be written if SEM or TEM,

Generally, there are no scalebars for images , and it is difficult to guess the size of the particles from micrographs

The part Material should be re-arranged , there should be written systematically;

This sentence I dont understand >The homogenous 115 mixture of these powders was prepared using planetary mill Pulverisette-6 (Fritsch 116 GmbH, Germany) with dry method in isopropyl alcohol during 2 hours. .... Dry in alcohol? 

Fig 4 ... the statement "pure' should be replaced for simulated or standard (if measured) , because your experimental sample is also pure.

Peaks should be noted with d or hkl.

Why authors didnt prepare images of SiO2 by themself?

Fig 6 - explanation of the M C Z means in caption of figure.

Where the .... zirconia, including its monoclinic phase.... is detected and proven?

for the EDS  spectra the part with Pt could be reduced to see the analyzed elements of the sample.

  This is sentence that has more errors than correct facts. "Figure 10 in spectrum 5, and the Table 3 demonstrated that there were 249 mainly tetragonal phase of zirconia, but monoclinic phase is present, too."

Presented manuscript could be very interesting , but it is containing so many  errors and wrong statements that it is highly lowering the scientific value.

Must be corrected and all graphical presentation done , so it will be clear for reader (not only for authors).

Author Response

Dear Editors and Reviewers,

Thank you very much for the thorough review and valuable suggestions. We addressed them all, and marked in red all the alterations in the revised paper. Please find detailed responses in the attached file.

Best regards

Reviewer 3 Report

The manuscript reports a study concerning the mechanical properties of ceramic composites in the system Al2O3-SiO2-ZrO2 sintered with a patented process based on a very high alternating current (3000 A) circulating in the graphite mold. The results conveyed by the manuscript appear interesting showing an improvement of mechanical properties compared to materials sintered with conventional process, but at the same time there are several ambiguous and unclear points.

Globally, the manuscript can be published in Materials upon clarification of these points:

-       - Firstly, the English of the manuscript should be improved;

-          - In lines 143-147, where electroconsolidation is described it is not clear the pressure of the process; in line 143 it is affirmed that electroconsolitation was carried out at 10-2 Pa, whereas in line 146 it is affirmed that better results were obtained at 35 MPa. Please clarify;

-          -The description of phase equilibrium (Section N.3, Theoretical background) is a little bit unclear, especially the linking of this section to the findings of the present work; please clarify; in addition, I suggest to refer also to updated literature in phase diagram of system Al2O3-SiO2-ZrO2 (doi: 10.1134/S1087659621050175).

-          - It is not clear how the data of conventionally sintered samples in Figure 12, 13, 14 and 15 were obtained. Did authors sinter also in conventional way these samples? If so, what were the conditions of the conventional sintering process? Why were not reported diffraction data of these samples? If not, what is the source of the data of conventionally sintered samples in these Figures?

Author Response

(The authors gave the same response as above.)

Round 2

Reviewer 1 Report

The author has revised the manuscript, however, English language should be polished before the manuscript is published. 

Author Response

Thank you for the positive review. The English style was polished, as recommended, with "track changes" option.

Best regards

Reviewer 2 Report

Dear authors , 

I can see that you were trying your best to improve your manuscript, very good. I can also understand now the Peculiarities, reading it for second time it is getting more peculiar. 

I have doubts . 

1.   fig 1 .... I am not sure  if this is SEM, I feel that is TEM. . . 

"SEM images of ZrO2– 5 wt.% Y2O3 nanopowders made by NANOE(France). " This formulation in English means that Image was made by Nanoe. I believe that it was purchased from that company.

2. However more relevant information  related to the image itself would be useful . There is no info about detection  of electrons (SE, BSE),  why the images are so different ( left is very bright , right is dull). And I got lost , Fig 1 is mixture of all ??? Then from the powder that is 50 microns big became  500nm powder ? just in 2 h? 

3. Since the size of the powder is very important it should be addressed in different way. e.g. particle size distribution PSD laser/ light , image analysis from SEM , LM. 

4. Regarding the XRD> statement : "From the XRD graphs shown in Figure 4 it can be concluded, that the powder contained some γ– and θ–Al2O3, causing difference from reference curve 2 made  for standard α–Al2O3 (99.5% pure). "  XRD pattern must have note where the other phase peaks are positioned , the PDF card numbers for all phases should be given , only than the analysis is trustful and can be checked. 

5. Elemental Composition and Microstructure chapter contains XRD results - this is phase analysis , no element can be evaluated. And here also the PDF cards are needed.

6. Figure 7. Elemental distribution in point 1 of the sample #4.... is not distribution , it is simple analysis at point. If authors are not willing modify the spectrum , they should et least write units for the axis.

7. From the SEM fig 7 - 10 , samples are very inhomogenous , there are visible big white particles, Explain this.

- EDS mullite 2Al2O3 SiO2  and corundum Al2O3 , how it is possible to separate? Explain and add to article. 

8. In my previous report I said: Where the .... zirconia, including its monoclinic phase.... is detected and proven? Response: It was shown in Fig. 6, peaks marked Z – zirconia (ZrO2 tetragonal), and Zm – zirconia (ZrO2 monoclinic).

Here you are stating that  Figure 10 in spectrum 4, and the Table 3 demonstrated that  there were mainly tetragonal phase of zirconia, but diffractograms in Fig. 6 indicated the  presence of monoclinic phase, too.

Please explain how  can you evaluate from elemental analysis tetragonal vs monoclinic phase?

This observation confirmed the results shown in Figure 4.  ..... in the figure 4 you present Al2O3.

Author Response

Dear Reviewer,

Thank you very much for the thorough review, which helped much to improve the quality of the paper. Please find attached the file with detailed responses, and you can see alterations in text made with “track changes” option.

Best regards
